# "I sometimes feel like I can't win!": An exploratory mixed-methods study of women's body image and experiences of exercising in gym settings

Emma S. Cowley [1,2] *, Jekaterina Schneider [3]

1 School of Sport and Exercise Science, Liverpool John Moores University, Liverpool, United Kingdom,
2 Department of Sport and Health Sciences, SHE Research Centre, Technological University of the Shannon: Midlands Campus, Athlone, Ireland, 3 Centre for Appearance Research, School of Social Sciences, College of Health, Science and Society, University of the West of England, Bristol, United Kingdom

* E.S.Cowley@ljmu.ac.uk

## Abstract

Despite an increase in gym memberships, women are less active than men and little is known about the barriers women face when navigating gym spaces. This study explored women's body image and experiences exercising in gyms. Two-hundred and seventy-nine women (84% current gym-goers; 68.1% White) completed an online mixed-methods survey. Thematic analysis of qualitative data produced four themes and nine subthemes: (1) "Never enough" ([perceived] judgement about appearance, [perceived] judgement about performance); (2) "Often too much" (self-criticism, clothing challenges); (3) "Always on display" (harassment and safety, fighting for space, the appearance contingency); and (4) "Sometimes empowered" (empowering places, empowering self). Qualitative and quantitative data showed that women often feel judged for their appearance and performance, leading to a persistent sense of inadequacy, as well as having to fight for space in the gym and to be taken seriously, while navigating harassment and unsolicited comments from men. Despite this, women showed signs of resistance towards gendered and appearance ideals permeating gym environments and some experienced empowerment through skill acquisition, breaking gender norms, and exercising in supportive environments. Based on our findings, we suggest a multi-level approach to tackling these barriers and creating more inclusive gym spaces for women.

**Data Availability Statement:** Quantitative data files are available from the Figshare database (via: doi. org/10.6084/m9.figshare.25959367.v1).

## 1. Introduction

Exercise significantly improves physical [1], mental [2], and psychosocial [3] health. Recent research indicates that women who engage in regular exercise experience greater health benefits than men, including lower incidence of all-cause mortality and reduced risk of cardiovascular events [4]. Exercise also alleviates postpartum depression symptoms [5], supports bone mineral density during postmenopause [6], and mitigates premenstrual syndrome symptoms

**Funding:** The author(s) received no specific funding for this work.

**Competing interests:** The authors have declared that no competing interests exist.

[7]. Specifically, resistance training enhances cognitive functioning [8], vitality [9], confidence, self-efficacy, and body image in women [10]. Despite these substantial benefits, many women do not achieve the recommended levels of physical activity [11]. According to recent World Health Organization (WHO) data, twenty to forty percent of women fail to meet the recommended minimum of 150 minutes of moderate-to-vigorous physical activity weekly, including muscle-strengthening activities on at least two days [12]. Furthermore, gender comparisons reveal that women are consistently less active than men throughout their lives [11, 13].

Studies show that women are more affected by body dissatisfaction and negative body image compared to men [14, 15]. Poor body image negatively impacts women's health behaviours and overall well-being, including feelings of low self-worth [16], higher incidences of depression [17], and increased risk of body dysmorphic disorders [18]. Psychological factors significantly influence women's decision-making regarding exercise [19], with poor body image predicting exercise avoidance [20].

Exercise motives and types vary based on body image perceptions. Appearance-focused exercise, like aerobics, is associated with more negative body image and self-objectification, while functionality-focused exercise, like resistance training, is associated with positive body image and empowerment [21]. Long-term female gym-goers report that resistance exercise provides a sense of accomplishment, boosts energy levels, and serves as a coping mechanism for managing emotions, with some likening gym experiences to "therapy" [22]. However, engagement in resistance exercise is complicated by gendered body ideals that valorise women as "thin and sexy", and men as "muscular and dominant" [23]. Even women who engage in resistance training often conform to these gendered stereotypes by lifting lighter weights to avoid becoming "too" muscular or using specific equipment to exercise certain body parts aligned with the ideal female physique [24].

#Fitspiration content on social media, designed to motivate people to exercise, has increased tenfold in the last decade [25]. This surge, along with the expansion of low-cost gym chains, has resulted in more women having gym memberships and incorporating regular exercise into their routines [26]. For some women, exercising in a gym fosters greater self-acceptance and personal growth [27], a sense of belonging within a supportive community [10], and empowerment through challenging gender norms [24].

Despite increased participation, gyms remain dominated by conventional masculine norms, creating significant barriers to equitable access for women [10], especially those who do not conform to socially constructed gender ideals and stereotypes [28]. The implicit gender segregation of the gym floor is a well-documented phenomenon [29], often causing women to feel intimidated and like "intruders" in the weights room [19]. In this area, men typically occupy more space, prioritise their need for equipment over that of women, and assert dominance through loud "grunting and groaning" noises [24]. Negative interactions with others, usually men, constitute another commonly cited barrier to positive gym experiences for women. Men often give unsolicited advice on women's technique and performance, along with negative comments and insults, particularly toward women whom they perceive as a threat [24]. Groups of men, described as "hyenas" for their objectification of women's bodies, frequently engage in unwanted advances, causing some women to stop exercising and leave the gym [19].

Women often incur additional labour and expenses to "fit in" and avoid unwanted attention. #Fitspiration emphasises women's appearance in gym settings, with some women spending up to 40 minutes applying make-up before workouts [19]. The gym layout, including mirrors and closely packed gym equipment, exacerbates body comparisons, leading to feelings of self-consciousness and inferiority [30]. Gym attire also serves as a marker of legitimacy, indicating experience level based on the brands worn [19]. By conforming to the gender

expectation of wearing form-fitting clothing, women aim to blend in and avoid appearing out of place [30].

Despite these challenges, gendered body and appearance stereotypes remain, and little is known about women's diverse gym experiences. Understanding these experiences is crucial for identifying and addressing barriers to access, thereby improving women's health and well-being. This study aimed to examine (i) the relationship between women's body image and gym experiences and (ii) the factors influencing women's experiences and behaviours in gym settings.

## 2. Materials and methods

### 2.1 Study design

This study implemented an online mixed-methods survey. Data collection took place in March 2024, and ethical approval was granted by the CONCEALED University Research Ethics Committee (ref no. 24/SPS/008).

### 2.2 Participants

Eligible participants were self-identified women, ≥18 years old, and current or former gym-goers. The study aimed to explore the experiences of both active and inactive women to identify the unique barriers and motivators influencing gym attendance and behaviours. Including both groups allowed for a comprehensive understanding of the factors that sustain regular gym participation and those that contribute to dropout. The survey was conducted in English and was open globally. Participants were recruited via social media advertisements with no incentives provided. Participants received an information sheet and completed screening questions to ensure eligibility, followed by voluntary informed consent before proceeding to the main survey.

### 2.3 Procedures

Participants answered a 130-item mixed-methods survey, hosted via Qualtrics (www.qualtrics.com), which took approximately 25 minutes to complete (S1 Appendix). Survey development was informed by existing literature [19, 31, 32], and the survey was piloted before data collection.

### 2.4 Measures

**2.4.1 Body image.** *2.4.1.1 Body Appreciation*. The 10-item Body Appreciation Scale (BAS-2) measured participants' body appreciation [33]. Statements like "I feel love for my body" were scored on a five-point Likert scale (1 = *Never*, 5 = *Always*), with higher scores indicating greater body appreciation. The BAS-2 showed good reliability and validity previously [34] and in the current study (α = .924).

*2.4.1.2 Functionality Appreciation*. The seven-item Functionality Appreciation Scale (FAS) assessed appreciation for bodies' capabilities [35]. Statements like "I appreciate my body for what it is capable of doing" were scored on a five-point Likert scale (1 = *Strongly disagree*, 5 = *Strongly agree*), with higher scores indicating greater functionality appreciation. The FAS showed good reliability and validity previously [36] and in the current study (α = .882).

*2.4.1.3 Sociocultural Attitudes Towards Appearance*. The Sociocultural Attitudes Towards Appearance Questionnaire (SATAQ-4R) [37] measured participants' thinness/low body fat (SATAQ-TL) and muscularity/athleticism internalisation (SATAQ-MA). Each component was measured by a separate five-item subscale that included statements like "I want my body

to look very thin" and "It is important for me to look athletic", scored on a five-point Likert scale (1 = *Definitely disagree*, 5 = *Definitely agree*), with higher scores indicating greater internalisation. The SATAQ-4R showed good reliability and validity previously [38] and in the current study (SATAQ-TL α = .861, SATAQ-MA α = .897).

*2.4.1.4 Internalised Weight Bias.* The 11-item Modified Weight Bias Internalisation Scale (WBIS-M) measured the degree to which participants apply weight-based stereotypes to themselves and base their self-evaluation on their weight [39]. Statements like "I hate myself for my weight" were scored on a seven-point Likert scale (1 = *Strongly disagree*, 7 = *Strongly agree*), with higher scores indicating greater internalised weight bias. The WBIS-M showed good reliability and validity previously [40] and in the current study (α = .938) (see Table 1).

**2.4.2 Gym behaviours.** Data regarding participants' gym behaviours were collected using open and closed questions across four categories.

*2.4.2.1 Physical Appearance.* Questions asked if factors such as sweat marks, skin concerns (e.g., acne, stretch marks, eczema), unshaved legs and underarms, and make-up use impacted women's gym experience. Participants also noted body parts they felt most and least confident about and concerns about becoming "too muscular".

*2.4.2.2 Gym Attire.* Questions explored preferences for gym clothing, comfort levels with shorts and crop tops, features prioritised in leggings and sports bras, and attire changes during menstruation.

*2.4.2.3 Gym Environment.* Questions covered preferred gym types (e.g., commercial or sport-specific gyms), gym habits (e.g., exercise times, exercising alone or with others), preferred gym areas (e.g., weights area, studio space, cardio machines), and feelings about exercising in front of mirrors.

*2.4.2.4 Interactions with Others.* Questions addressed feelings of intimidation and/or judgement in the gym, unsolicited comments, and experiences of physical and/or sexual harassment.

**2.4.3 Demographics.** Participants provided information on age, country, ethnicity, disability, sexual orientation, frequency and duration of weekly physical activity, predominant physical activity type, perceived weight, and current weight goals.

## 2.5 Data analysis

**2.5.1 Quantitative analysis.** Statistical analyses were conducted using SPSS v. 27.0 (SPSS, Inc., Chicago, IL). Data were examined for outliers, skewness, and kurtosis; no outliers were found, and data were normally distributed (skewness and kurtosis ≤+/-2.58). Pearson's correlations determined associations between psychosocial variables [41] where $r = 0.10$–$0.30$, $r = 0.30$–$0.50$, and $r \geq 0.50$ represent a small, medium, and strong effect, respectively. Differences in psychosocial variables between past and current gym-goers were assessed using independent *t*-tests, with significance set at *p*-value < .05. Gym behaviour data (open and closed

**Table 1. Descriptives of psychosocial measures.**

| | Range | Mean | SD | Skewness | Kurtosis | Alpha (α) |
|---|---|---|---|---|---|---|
| Body appreciation | 1–5 | 3.45 | 0.68 | -.102 | -.283 | .924 |
| Functionality appreciation | 1–5 | 4.38 | 0.51 | -.802 | 1.152 | .882 |
| Muscularity/ athleticism internalisation | 1–5 | 3.42 | 0.96 | -.314 | -.485 | .897 |
| Thinness/ low body fat internalisation | 1–5 | 2.92 | 0.99 | .044 | -.616 | .861 |
| Weight bias internalisation | 1–7 | 3.00 | 1.41 | .774 | -.240 | .938 |

questions) were analysed together and presented within themes. Descriptive statistics are presented alongside the themes to support the findings from the qualitative data.

**2.5.2 Qualitative analysis.** Open-ended questions were analysed using reflexive thematic analysis [42, 43]. This approach is particularly suited to exploring nuanced, subjective experiences, fitting well within a constructionist epistemology [43]. The first author (a researcher exploring physical activity, behaviour change, and gender data bias) led the analysis with support from the second author (a researcher conducting applied body image, weight bias, and weight stigma research, focused on sport and exercise settings). Both authors identify as White European women. The first author became familiar with the data by cleaning and organising the dataset and generated initial themes and subthemes by clustering participant quotes that had similar meaning. Themes were initially deductively guided by the survey's four "gym behaviours" categories, and subsequently analysed inductively using the author's understanding of feminist theory and existing literature. The authors had regular meetings to discuss, debate, and refine the thematic structure, resulting in multiple iterations of the core themes, subthemes, and selected quotes. Themes are presented narratively below, alongside supporting quotes. To ensure anonymity, participants are presented with an identifier (e.g., P1) and their age (see Fig 1 for thematic map).

This study adopted a relativist ontological stance. Relativism posits that reality is not singular or objective but is constructed and interpreted differently by various individuals or groups [42, 44]. This perspective aligns with reflexive thematic analysis, acknowledging that themes

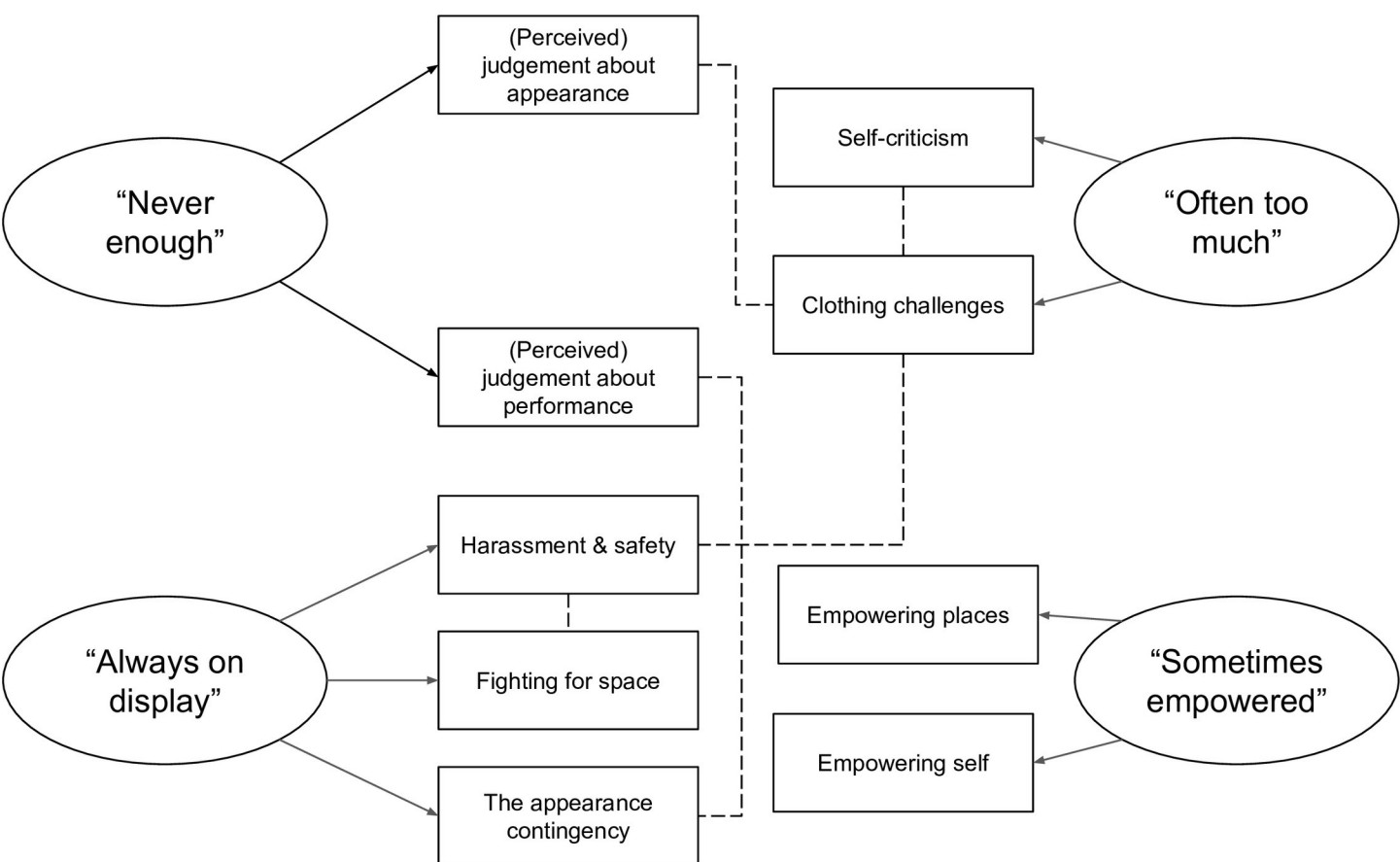

**Fig 1. Thematic map of core themes and subthemes.** *Note.* Solid lines with arrows represent subthemes. Dashed lines represent relationships between themes.

are actively created by the researcher through their interaction with the data (43). Our epistemological position is grounded in constructionism, which suggests that knowledge is constructed through social processes and interactions, rather than reflecting an objective reality [45, 46]. This aligns with the reflexive nature of thematic analysis, which emphasises the researcher's role in shaping the analysis through their own perspectives and experiences [44, 45]. The qualitative insights gained from the analysis were integrated with quantitative findings to provide a comprehensive understanding of the research problem. Throughout the analysis, reflexivity was a critical component [42]. We continuously reflected on how our own backgrounds, assumptions, and interactions with the data influenced our analysis.

# 3. Results

## 3.1 Participants

Participant recruitment yielded 420 responses. Responses were removed for not providing consent ($n = 96$), failing the screening questions ($n = 11$), providing duplicate responses ($n = 9$), failing the reCAPTCHA (Completely Automated Public Turing test to tell Computers and Humans Apart) test ($n = 21$), or being identified as fraudulent or bot based on Qualtrics data quality screening procedures and recently published guidelines for detecting fraudulent responses in web-based research [47] ($n = 4$), leaving 279 participants. The majority of participants were current gym-goers ($n = 235$; 84.2%), 30–39 years old ($n = 90$; 32.3%), White ($n = 190$; 68.1%), perceived themselves to be of "normal weight" ($n = 147$; 52.7%), were currently trying to lose weight ($n = 105$; 37.6%), exercised 2–3 times/week ($n = 134$; 48.0%), for 3–6 hours/week ($n = 98$; 35.1%), and predominantly engaged in strength training ($n = 153$; 54.8%) (see Table 2). No differences were found regarding psychosocial variables between current and past gym-goers, apart from on the muscularity/athleticism internalisation scale, with current gym-goers reporting a higher drive to look muscular/athletic ($M = 3.54$, $SD = 0.91$) than past gym-goers ($M = 2.72$, $SD = 0.97$), $t(248) = 5.009$, $p < .001$.

## 3.2 Correlations between psychosocial variables

The results of Pearson's correlations were significant and in the expected direction, with one exception. No significant correlations between muscularity/athleticism internalisation with body appreciation, functionality appreciation, or weight bias internalisation were found ($ps > .05$). However, muscularity/athleticism internalisation was positively associated with thinness/low body fat internalisation. Positive correlations between body and functionality appreciation, and negative correlations between body and functionality appreciation with thinness/low body fat internalisation and internalised weight bias were observed (see Table 3).

## 3.3 Theme 1. "never enough"

Thematic analysis of qualitative data resulted in four core themes and nine subthemes. Regarding the first theme, many women reported feeling judged both for their *appearance* and *performance* in the gym. Although women said that they would not judge others, they still worried about being judged: "I worry that other people will judge me or laugh at me for being an overweight person attending the gym" (P247, 30–39) and "I would never have an issue with another woman's body hair, but still feel self-conscious about my own" (P20, 30–39). This perceived judgement resulted in feelings of never being "enough".

 **3.3.1 (Perceived) judgement about appearance.** Appearance-related concerns were varied and complex, encompassing not only feelings of self-consciousness regarding physical body shape and size, but also discomfort related to hair, skin issues, and other concerns.

**Table 2.** Demographics of study sample.

| | Full Sample (*N* = 279) | Current Gym-Goers (*N* = 235) | Past Gym-Goers (*N* = 44) |
|---|---|---|---|
| | *N*(%) | *N*(%) | *N*(%) |
| Age (years) | | | |
| <25 | 25(9.0%) | 24(10.2%) | 1(2.3%) |
| 25–29 | 55(19.7%) | 49(20.9%) | 6(13.6%) |
| 30–39 | 90(32.3%) | 74(31.5%) | 16(36.4%) |
| 40–49 | 34(12.2%) | 27(11.5%) | 7(15.9%) |
| 50–59 | 6(2.2%) | 4(1.7%) | 2(4.5%) |
| 60+ | 2(0.7%) | 2(0.9%) | |
| Missing | 67(24.0%) | 55(23.4%) | 12(27.3%) |
| Country | | | |
| United Kingdom | 97(34.8%) | 84(35.7%) | 13(29.5%) |
| United States | 58(20.8%) | 52(22.1%) | 6(13.6%) |
| Netherlands | 2(0.7%) | 2(0.9%) | |
| Switzerland | 1(0.4%) | 1(0.4%) | |
| Sweden | 5(1.8%) | 3(1.3%) | 2(4.5%) |
| Slovakia | 1(0.4%) | 1(0.4%) | |
| Ireland | 24(8.6%) | 17(7.2%) | 7(15.9%) |
| Germany | 1(0.4%) | 1(0.4%) | |
| Finland | 2(0.7%) | 2(0.9%) | |
| Denmark | 2(0.7%) | 2(0.9%) | |
| Canada | 8(2.9%) | 5(2.1%) | 3(6.8%) |
| Belgium | 1(0.4%) | 1(0.4%) | |
| Australia | 9(3.2%) | 8(3.4%) | 1(2.3%) |
| Missing | 68(24.4%) | 56(23.8%) | 12(27.3%) |
| Ethnicity | | | |
| Asian or Asian British/ Irish | 7(2.5%) | 4(1.7%) | 3(6.8%) |
| Mixed/ multiple ethnic groups | 9(3.2%) | 6(2.6%) | 3(6.8%) |
| White/ Caucasian | 190(68.1%) | 165(70.2%) | 25(56.8%) |
| Other | 6(2.2%) | 5(2.1%) | 1(2.3%) |
| Missing | 67(24.0%) | 55(23.4%) | 12(27.3%) |
| Disability | | | |
| Yes | 11(4.0%) | 9(3.8%) | 2(4.5%) |
| No | 201(72.0%) | 171(72.8%) | 30(68.2%) |
| Missing | 67(24.0%) | 55(23.4%) | 12(27.3%) |
| Sexual orientation | | | |
| Asexual | 5(1.8%) | 5(2.1%) | |
| Bisexual | 20(7.2%) | 19(8.1%) | 1(2.3%) |
| Lesbian | 8(2.9%) | 8(3.4%) | |
| Queer | 2(0.7%) | 2(0.9%) | |
| Heterosexual | 172(61.6%) | 142(60.4%) | 30(68.2%) |
| Other | 1(0.4%) | 1(0.4%) | |
| Prefer not to say | 4(1.4%) | 3(1.3%) | 1(2.3%) |
| Missing | 67(24.0%) | 55(23.4%) | 12(27.3%) |
| Weight perception | | | |
| "Underweight" | 2(0.7%) | 2(0.9%) | |
| "Normal weight" | 147(52.7%) | 126(53.6%) | 21(47.7%) |
| "Overweight" | 62(22.2%) | 51(21.7%) | 11(25.0%) |

*(Continued)*

**Table 2.** (Continued)

| | Full Sample (*N* = 279) | Current Gym-Goers (*N* = 235) | Past Gym-Goers (*N* = 44) |
|---|---|---|---|
| | *N*(%) | *N*(%) | *N*(%) |
| Prefer not to say | 1(0.4%) | 1(0.4%) | |
| Missing | 67(24.0%) | 55(23.4%) | 12(27.3%) |
| **Weight change behaviours** | | | |
| Lose weight | 105(37.6%) | 90(38.3%) | 15(34.1%) |
| Gain weight | 7(2.5%) | 6(2.6%) | 1(2.3%) |
| No | 97(34.8%) | 82(34.9%) | 15(34.1%) |
| Prefer not to say | 3(1.1%) | 2(0.9%) | 1(2.3%) |
| Missing | 67(24.0%) | 55(23.4%) | 12(27.3%) |
| **Gym frequency/week** | | | |
| Once | 27(9.7%) | 20(8.5%) | 7(15.9%) |
| 2–3 times | 134(48.0%) | 112(47.7%) | 22(50.0%) |
| 4–6 times | 105(37.6%) | 98(41.7%) | 7(15.9%) |
| Every day | 5(1.8%) | 5(2.1%) | |
| Prefer not to say | 8(2.9%) | | 8(18.2%) |
| **Exercise frequency/week (hours)** | | | |
| <1 | 4(1.4%) | 3(1.3%) | 1(2.3%) |
| 1–3 | 36(12.9%) | 24(10.2%) | 12(27.3%) |
| 3–6 | 98(35.1%) | 84(35.7%) | 14(31.8%) |
| 7–9 | 90(32.3%) | 80(34.0%) | 10(22.7%) |
| 10+ | 51(18.3%) | 44(18.7%) | 7(15.9%) |
| **Exercise type** | | | |
| Cardio | 69(24.7%) | 50(21.3%) | 19(43.2%) |
| Strength | 153(54.8%) | 138(58.7%) | 15(34.1%) |
| Flexibility | 10(3.6%) | 6(2.6%) | 4(9.1%) |
| Other | 47(16.8%) | 41(17.4%) | 6(13.6%) |

> For me, I've always been comfortable training in the gym without make-up or hair done. But when I struggle with my acne it can impact. In the past, I've gone to the gym and then turned back around because I was so anxious about my skin. I've also worn pimple patches, or concealer to hide acne, which makes me feel uncomfy and sad (I hate sweating with make-up on my face!!). I have thought about why I do this just to make others feel comfortable. . . (P180, 25–29)

**Table 3. Pearson's correlations between psychosocial variables.**

| | 1. Body appreciation | 2. Functionality appreciation | 3. Muscularity/ athleticism internalisation | 4. Thinness/ low body fat internalisation | 5. Weight bias internalisation |
|---|---|---|---|---|---|
| 1. Body appreciation | | | | | |
| 2. Functionality appreciation | .626** | | | | |
| 3. Muscularity/ athleticism internalisation | -.016 | .056 | | | |
| 4. Thinness/ low body fat internalisation | -.327** | -.216** | .394** | | |
| 5. Weight bias internalisation | -.745** | -.527** | .039 | .370** | |

**p < .001.

Women reported feeling self-conscious about how they looked in the gym and feared judgement for wearing clothes that might reveal or highlight "problem areas". This fear of judgement was observed across body types, as even women who align more closely with societal ideals (e.g., thin, slender, feminine) expressed feeling inadequate or the need to hide certain body parts to avoid judgement. For example, women said they felt self-conscious about "looking manly", being "flat chested", having "big boobs" or "large breasts", or being too "fat", "wobbly", "muscular", "small", "lean", or "easy".

> I naturally gain muscle really easily (unlike many women) and sometimes I worry about how having muscle is perceived. Diet, sleep, and exercise is incredibly important to me, and sometimes I worry that people assume other more negative methods. Sometimes I avoid wearing tank tops despite being proud of my arm muscles. (P168, 25–29)

However, most women in the current sample were not concerned about becoming "too" muscular ($n = 206$; 87.3%).

Several women shared their distress about conflicting appearance ideals for women within and outside of the gym ("I want to be lean and muscular but not too broad because I still want to be small and thin—it's so distressing" P60, 25–29) and the societal pressures for women to look "sexy" and "small". This led to feeling their appearance is "never enough" and that women simply "can't win".

> In general, the clothes [I wear] to the gym are quite modest (e.g., loose T-shirt and leggings). I don't want to succumb to the pressure of wearing revealing clothes in the gym like other women do, but then I feel self-conscious that I don't look "sexy" enough. On occasions where I've worn shorts to the gym, I then feel self-conscious as I'm aware of people (especially men) looking at me—I sometimes feel like I can't win! (P81, 30–39)

**3.3.2 (Perceived) judgement about performance.** Women similarly felt that others would judge them for their form, technique, or skills in the gym or make assumptions about their fitness, knowledge, and ability. This was expressed through fears of looking "stupid" (P247, 30–39), "like a newbie" (P178, 30–39), "like a fool" (P112, 25–29), like "you don't know what you're doing" (P74, 25–29), or thinking other people will "judge [their] form" (P171, 25–29). To prevent this, women felt pressure to wear specific and matching gym attire ("Sometimes I want to wear my work pants, which are like quick-dry cargo pants, but I think I look like a newbie if I don't wear leggings" P178, 30–39), while at the same time not looking too dressed up for fear of not being taken seriously ("I think wearing new clothing sets and looking "too" put together may make others take me less seriously" P7, 30–39). In this way, women highlighted the core theme ("never enough") through underlying nuances of not being able to win regardless of what they do or wear.

A key aspect of women's perceptions that others would judge their performance was linked to the idea that fitness has a particular "look" and that women whose bodies do not align with this look would be perceived as novice exercisers and be more likely to be form-corrected in the gym ("[. . .] my physical appearance does not reflect my fitness" P254, 40–49; "I'm small and don't look like I can lift so sometimes am not taken seriously" P143, 40–49).

> My body doesn't reflect my athletic and physical abilities. I'm short and have an above average BMI for my age/height. However, I exercise more than anyone I know. I compete at a high level of sport and am very active in my job. I eat well and don't over-indulge and tend

to track things that I eat. WHY does BMI make me feel bad even though my body's abilities don't reflect what the numbers say? (P150, <25)

### 3.4 Theme 2. "often too much"

Women were critical of themselves and their own bodies and used words such as "excessive", "extra", and "too" to convey this criticism ("Stomach too big" P128, 30–39; "Too much gut fat" P173, 40–49; "My legs are too fat" P100, 30–39; "I carry extra belly fat on my frame" P11, 40–49; "I feel too old for [crop tops]" P125, 40–49; "Excessive sweat" P178, 30–39). This was partially linked to limited gym attire options and sizes and women's discomfort with how their bodies look or feel in particular clothing.

**3.4.1 Self-criticism.** Women shared their concerns about many different body parts, with the stomach/belly area evoking the most criticism ("I hate my stomach" P73, 25–29; "I'm too old, and my waist/stomach does not look good in a crop top" P239, 40–49). Indeed, 55.0% of women ($n$ = 131) reported that their stomach is the body part they feel *least* confident about, followed by legs ($n$ = 31; 13.0%) and arms ($n$ = 23; 9.7%). In contrast, the stomach was rated as second to last in terms of the body parts women were *most* confident about ($n$ = 7; 2.9%), with only the chest being rated lower ($n$ = 5; 2.1%), and legs ($n$ = 53; 22.3%) and bum ($n$ = 45; 18.9%) rated most highly. However, a wide range of body parts and features were criticised among women in the current sample, including "sweat", "hair growth", "facial hair", "bum", "thighs", and "skin". In terms of skin concerns, 60 women (25.0%) reported concerns about stretch marks; 46 (19.2%) about facial acne; 18 (7.5%) about spots on shoulders, back, or chest; 15 (6.3%) about eczema; and 30 (12.5%) about other concerns (e.g., scars, wrinkles, cellulite, redness, loose skin). Further, 30 women (17.8%) reported that skin concerns impacted how they felt about exercising in a gym.

**3.4.2 Clothing challenges.** Women often discussed criticisms of their bodies in combination with concerns about gym clothes. A recurrent concern was related to the groin area ("Sometimes if I'm wearing leggings or shorts, I get insecure about camel toe even when wearing appropriate underwear" P21, <25; "Oftentimes when I'm doing cardio, I'm worried of the sweat marks I get on my leggings, sometimes it looks like I've peed myself!" P110, <25; "Being able to see period knickers when wearing leggings" P197, age unreported). Women also reported struggles in finding clothes that fit comfortably.

> Being more muscular than your average gal, 99% of clothes aren't made for my body in mind and I'm very aware of that and it doesn't make me feel great. My legs don't fit in things that also fit my waist, I've minimal boob but wide lats and big shoulders/arms and being in globo gyms I feel very self-conscious as it emphasises how different my body is to others. (P136, 30–39)

Although most women found sufficient choice in activewear for their body shape and size ($n$ = 165; 72.7%), many had difficulty purchasing sports bras ($n$ = 62; 59.6%) and leggings ($n$ = 42; 40.4%). They were also concerned about showing underwear lines/seams when wearing leggings ($n$ = 109; 48.4%).

### 3.5 Theme 3. "always on display"

The third theme centred on women's bodies being on display, often for (male) consumption. This was reflected in experiences of harassment and safety concerns, a sense of not belonging, and internalised appearance ideals.

**3.5.1 Harassment and safety.** Women reported *past* ("In secondary school I once had a boy tell me I've tree trunk legs. It has stuck over 10 years later" P150, <25; "Male gym teacher commented I hadn't made the effort for his class as my nail varnish was chipped" P261, 30–39), *current* ("Male gaze, feel more comfortable covering my butt" P276, age unreported; "Men getting in my personal space when exercising" P176, 40–49), and *expected* harassment and safety concerns ("I don't want to be stared at" P82, 25–29; "I don't want to attract any unwanted attention from men" P152, 30–39). These concerns were predominantly perpetuated by men, and women described feeling "sexualised", "annoyed", "objectified", "uncomfortable", "anxious", "on the defense", "creeped out", "uneasy", "scared", "angry", "unsafe", "exposed", "threatened", "gross", "sad", "unworthy", "horrible", and "self-conscious" due to unsolicited stares and comments from men. When asked to describe how it would feel if a woman or man stared at them in the gym, most women chose "curious" for a woman ($n = 89$; 42.2%) and "intimidated" for a man ($n = 58$; 27.5%). Eighty-three women (39.2%) reported being more intimidated by men when exercising, while 62 (29.2%) felt more judged by men.

> I generally don't want to be approached by anyone at the gym unless they are a friend. A female friend of mine was murdered by a man with whom we both went to the gym. This probably contributes to me feeling unsafe/uneasy interacting with men at the gym. I'm less afraid when my husband and I go together. We don't exercise together, but I feel better knowing that he is in the same building. (P237, 30–39)

When asked if they consider potential dangers (e.g., unwanted attention, comments, or physical touch) when deciding what to wear to the gym, 41.4% of women said 'yes' ($n = 79$) or 'unsure' ($n = 12$). This appears warranted as 46.6% ($n = 89$) reported receiving unsolicited compliments on their appearance at the gym, mostly from men ($n = 64$; 71.9%).

**3.5.2 Fighting for space.** Women reported having to "fight" for space in the gym. They felt like they did not belong and should not take up "too much space" (P178, 30–39), whereas men took up an "unholy amount of space" (P110, <25) and felt entitled to take gym equipment women were using. When women took up space, they faced unwanted stares, comments, and criticism from men ("I weighed about 130lbs and was using a leg press machine and a much larger man told me 'get off the machine string bean, you're gonna hurt yourself'" P24, 25–29; "He suggested I use the women's only area" P145, 30–39).

> I've frequently been the only woman squatting or deadlifting where my rack only has one or two 45lb plates left hanging up. Several times I've had men come and take the 45lb plates from my squat rack even though other racks were closer to them or had more plates available. This often made me upset as there's no reason for them to take the weight from my rack other than they don't want to take it from a man. (P24, 25–29)

**3.5.3 The appearance contingency.** Several women conveyed their internalised appearance ideals. For example, although most women felt self-conscious wearing certain items of clothing, others felt comfortable wearing shorts, sports bras, and crop tops in the gym ("I feel comfortable wearing shorts to the gym because my legs are moderately muscular" P15, <25). Notably, this was largely contingent on them feeling comfortable in, or liking the appearance of, their bodies. Some women explicitly admitted that their comfort levels of wearing certain clothes were likely tied to their bodies being "thin" and meeting societal appearance ideals.

> As much as I'd love to say it's because I don't care about how I look and when it's hot I feel more comfortable in [shorts], I also know by societal norms, I have a "good" body and "good" legs and they're a part of my body I've never felt insecure about. (P141, 25–29)

Others implicitly hinted at this, describing their bodies with phrases aligned with appearance ideals, such as "I have a defined stomach so I feel confident" (P124, 30–39), "I'm fairly slim so feel comfortable in a crop top" (P142, 25–29), "I have a big butt (P127, 30–39)", "I have always had a flat stomach" (P111, 30–39), etc. This theme is related to the idea that fitness has a certain "look" and that women fitting this look feel more comfortable wearing specific gym attire. In other words, comfort and belonging in the gym are contingent on appearance.

### 3.6 Theme 4. "sometimes empowered"

The final theme relays women's positive experiences of feeling empowered in the gym, particularly when they defy gender norms (e.g., lift heavy weights, outlift men) or exercise in safe and inclusive spaces. For some women, the feeling of safety and empowerment came from changing gyms, while for others, it came from gaining skills and experience.

**3.6.1 Empowering places.** Women felt safer and more empowered in "inclusive", "respectful", "positive", "friendly", and "encouraging" gyms, where performance was prioritised over appearance. These were often private or CrossFit gyms, rather than commercial or public gyms. Women-only spaces also made women feel comfortable, confident, and empowered.

> I prefer working out in my small gym, which is focused more on strongman/Olympic style lifting than in a general-purpose commercial gym because there's much less judgement and people seem to be friendlier. (P49, 40–49)

> Now that I participate in hot yoga with weights, I'm in an environment where there are mostly females and I think it's contributed to why I don't care about a lot of these things anymore. (P238, <25)

However, most women still preferred commercial chain gyms ($n = 78$; 35.5%), followed by personal training or small group training gyms ($n = 37$; 16.8%), and typically exercised alone ($n = 137$; 62.3%), suggesting that other factors more strongly influenced the type of gym women selected to exercise in.

Regarding exercising in front of mirrors, 19 women (8.7%) said that they hated it, 34 (15.6%) preferred not to, 61 (28.0%) preferred to, 23 (10.6%) loved it, and 81 (37.2%) said they did not care. Women who liked exercising in front of mirrors primarily cited checking form as the reason for doing so, while women who did not cited feeling self-conscious about their appearance and being distracted.

**3.6.2 Empowering self.** Several women gained confidence over time, feeling more comfortable going to the gym, using equipment, and taking up space. Some found it empowering to break gender stereotypes and norms by outlifting men ("Very empowering as a woman when you approach a male-dominated weight area, feel the judgemental stares from men wondering what is SHE doing here, and end up outlifting them all" P180, 25–29), lifting heavy weights ("Lifting weights makes me feel like a strong ass lady" P103, <25), or exercising in traditionally male-dominated gym spaces.

> I've always found it so fascinating how men respond to women in the free weights area, especially strong women. It is not without challenge, but I also think it's extremely

empowering for women (i.e., to make room for yourself in a space dominated by men, at their testosterone peak). (P180, 25–29)

I used to be much more self-conscious at the gym, but after a year of training I feel like I deserve to be here just like anyone else and therefore I am less self-conscious. I am ok doing exercises I don't know how to do even in my sports bra in a full gym. (P91, 30–39)

It should be noted that in the current sample, most women felt most confident doing strength-based exercise ($n = 133$; 47.7%) and using the free weights area ($n = 118$; 54.6%), and least comfortable using resistance machines ($n = 52$; 24.2%), which may explain why they felt empowered in these spaces.

## 4. Discussion

This study explored women's body image and gym experiences related to their physical appearance, gym attire, gym environment, and interactions with others. Both qualitative and quantitative data highlighted challenges women face in the gym, revealing contradictions in how women attend the gym, train, and interact. One key finding was women's perceptions of gym attire as both a barrier and facilitator to exercise. Aligning with previous literature [19, 48], women often chose attire based on comfort and functionality. However, their choices were also influenced by comparisons with others (e.g., looking less "sexy" than other women) or fear of judgement for wearing non-branded attire (e.g., looking like a "novice") or looking "too put together" (e.g., not being taken seriously). Many women also chose gym attire to hide perceived "problem areas" or avoid appearance concerns, including visible sweat stains.

Additionally, contradictions arose in women's conflict between accepting diverse appearances, while also placing significant meaning on appearance. Women reported being non-judgemental and accepting of other women's bodies and attire but felt self-conscious about their own appearance. For women who felt less self-conscious in how they showed up to the gym, an appearance contingency was evident. Specifically, women implicitly or explicitly linked their comfort and confidence to whether their bodies aligned with societal appearance ideals. This was reflected by other women not feeling comfortable wearing shorts or crop tops or exercising in sports bras if they had concerns about their appearance or perceived themselves to be "too" fat. These findings suggest there is a "correct" way to look in the gym. Prior research has found that gym environments often reinforce the idea of a particular "look" associated with fitness [49], conflating fitness with health and neglecting the social determinants of health and the natural variation of the human body in shape, size, and ability [49]. The ideals within (e.g., muscular, lean, athletic) and outside the gym (e.g., thin, curvy) clash, creating pressures for women to meet both ideals simultaneously [48].

Notably, the current sample showed high muscularity/athleticism internalisation, with most women preferring strength-based exercise. Participants reported high body and functionality appreciation scores, but also high muscularity/athleticism internalisation, with current gym-goers reporting higher internalisation than former gym-goers. This finding may be linked to shifting appearance ideals, specifically the idea that "strong" or "fit" is "the new skinny" [50]. The rise of the muscularity ideal among women appears to reflect a broader shift in appearance ideals from thinness to fitness, often characterised by lean, toned, and muscular physiques. This trend is influenced by the proliferation of social media content, particularly #Fitspiration, which promotes "strong" and "fit" as desirable traits. This cultural shift could explain the higher internalisation of muscularity and athleticism ideals observed in our sample. Although the "fit ideal" was previously considered healthier and more attainable than the "thin ideal", research shows that it similarly reinforces unattainable [51] and unhealthy [52] ideals.

Recent studies have found that #Fitspiration content highlighting the "strong" or "fit" ideal can worsen women's body image [50, 53] and that fit ideal internalisation is linked to poor body image and eating and exercise problems [54, 55]. Although we found no significant correlations between muscularity/athleticism internalisation with body appreciation, functionality appreciation, or weight bias internalisation, muscularity/athleticism internalisation was positively associated with thinness/low body fat internalisation, suggesting that women high in muscularity/athleticism internalisation also internalised the thin ideal.

Finally, contradictions were observed in how women responded to men's presence in the gym. While some found it empowering, many reported feeling unsafe, sexualised, and harassed. Indeed, 71.9% of women reported having received at least one unsolicited comment from a man in the gym. These findings align with previous research highlighting intimidation and harassment as common in public gym spaces [19]. Many commercial gyms remain gendered and dominated by men, with women feeling pressured to conform to feminine appearance ideals while fighting to take up space and prove their strength and abilities [56, 57]. Despite this, women in our sample and prior research [49] show signs of resistance towards the gendered and appearance ideals permeating gym environments. Participants highlighted inconsistencies and dilemmas in choosing what to wear and challenged their own thoughts about their bodies. Additionally, several women reported feeling empowered when occupying traditionally "male" spaces in the gym, such as the weights area, and outperforming men.

## 4.1 Limitations

The findings of the current study must be considered in light of several limitations. Despite the diverse geographical representation, most participants were White, heterosexual, identified as "normal weight" and able-bodied, were current gym-goers, and lived in western, educated, industrialised, rich, and democratic (WEIRD) countries. This limits the applicability of our findings to women who no longer engage in exercise (particularly in a gym setting) or those of diverse ethnicities, body sizes, sexualities, and abilities. Women in larger and marginalised bodies, particularly those with intersecting identities (e.g., women of colour, women with disabilities), are likely to face additional challenges within the gym environment, related to gym attire, accessibility, and interactions. Evidence suggests that ethnic and racial backgrounds influence how women perceive feminine and appearance ideals. Women of colour often embrace a "thicker" body ideal and adopt a more flexible perspective on the female body that is not centred on achieving a cultural body standard [58, 59]. Sexual orientation may also affect women's views on these ideals, with some studies suggesting that lesbian women may be less focused on the thin ideal compared to heterosexual women [60]; however, recent studies do not support this [61, 62]. Predominant engagement in strength training among our participants, coupled with high levels of muscularity/athleticism internalisation, may not reflect the experiences of all women. Future research should address these limitations through more inclusive sampling strategies, focusing on women of colour, women in larger bodies, and women with disabilities, as well as past gym-goers.

## 4.2 Implications

Building on our findings and prior research, we outline important implications for creating safe, inclusive, and empowering gym spaces for women at the individual, interpersonal, organisational, and societal levels (see Fig 2). For example, individuals should consider focusing on non-appearance related goals, emphasising the physical, mental, and emotional health benefits of exercise. To foster safer interpersonal relations, gyms should enforce comprehensive anti-harassment policies and provide staff training to cultivate supportive and inclusive

| | |
|---|---|
| ⇒ Individual Level | ✔ Implement interventions to improve women's body image and reduce the internalisation of muscularity and athleticism ideals, particularly for women currently engaged in strength training.<br>✔ Promote a holistic approach to fitness, encouraging women to exercise for a variety of reasons not attached to appearance (e.g., improving strength, enhancing mental health and well-being, fostering overall health). |
| ⇒ Interpersonal Level | ✔ Implement and enforce comprehensive anti-harassment policies within gyms. These policies should be clearly communicated to all members and staff, establishing a zero-tolerance stance on any form of harassment.<br>✔ Provide thorough training for gym staff on how to recognise and effectively address instances of harassment, as well as on how to foster a supportive and inclusive atmosphere.<br>✔ Establish robust support systems and clear reporting mechanisms (e.g., anonymous hotlines, dedicated email addresses, or in-person reporting options). These systems should ensure timely and appropriate responses to complaints. |
| ⇒ Organisational Level | ✔ Design gym layouts that are accessible and comfortable for women of all body sizes and types. This means ensuring that equipment is user-friendly and can accommodate a diverse range of needs, from height and weight adjustments to easy-to-use interfaces.<br>✔ Provide inductions and orientations for new members in safe spaces to ensure they feel comfortable using different areas and pieces of equipment.<br>✔ Emphasise communication and advertising strategies that decouple weight and body size from health and include diverse representation across gender, age, race/ethnicity, body size, and ability.<br>✔ Produce gym attire that considers women's needs in the gym and that fits a wide range of body shapes and sizes. |
| ⇒ Societal Level | ✔ Develop and promote industry standards that require gyms to provide equipment and facilities that accommodate a diverse range of body sizes and types.<br>✔ Foster cultural shifts in societal perceptions of fitness that emphasise the value of physical activity for health and well-being, rather than focusing solely on aesthetic goals. This can be achieved through comprehensive public health campaigns and positive media representation, which highlight the diverse benefits of exercise and celebrate all body types engaging in physical activity. |

**Fig 2. Recommendations for creating safe and inclusive gym spaces for women.**

environments. Gym layouts should be designed to prioritise accessibility, ensuring that spaces and equipment accommodate diverse needs and enhance users' comfort and confidence. Finally, public health campaigns should promote a cultural shift by celebrating exercise as a means to improve health and well-being, moving away from a sole focus on aesthetics.

## 4.3 Future directions

Future research should employ more targeted recruitment strategies to ensure the inclusion of underrepresented groups, such as women from non-WEIRD countries, those from the LGBTQIA+ community, and women with disabilities. Research should also explore the experiences of women who have discontinued gym-based exercise to gain a better understanding into the factors contributing to their drop-off.

Building on the findings of this study, follow-up research is also required to understand the contradictory results highlighted in the data. For example, the majority of participants reported feeling neutral or positive about exercising in front of mirrors, which runs contrary to much of the existing literature. Further, investigation is also warranted into the evolving muscularity ideals and the shifting perceptions of the 'ideal' female body, as these cultural factors likely influence women's experiences and engagement in gym settings. To this end, we are currently conducting a follow-up qualitative study to gain greater insights into these findings.

## 5. Conclusions

This study expands on prior research by highlighting the barriers women face when exercising in gyms across various domains, including physical appearance and body image, gym attire, the physical gym environment, and in interactions with others. The combined qualitative and quantitative data shed light on important contradictions that are experienced by women across these domains. Based on the findings of this research, we propose a multi-level approach to support women in accessing and feeling comfortable in gym spaces, to increase physical activity and reduce exercise disengagement among this population.

## Supporting information

**S1 Appendix. Study survey.**
(DOCX)

## Author Contributions

**Conceptualization:** Emma S. Cowley.

**Data curation:** Emma S. Cowley, Jekaterina Schneider.

**Formal analysis:** Emma S. Cowley, Jekaterina Schneider.

**Investigation:** Emma S. Cowley, Jekaterina Schneider.

**Methodology:** Emma S. Cowley, Jekaterina Schneider.

**Project administration:** Emma S. Cowley, Jekaterina Schneider.

**Resources:** Emma S. Cowley, Jekaterina Schneider.

**Software:** Jekaterina Schneider.

**Validation:** Emma S. Cowley, Jekaterina Schneider.

**Visualization:** Emma S. Cowley, Jekaterina Schneider.

**Writing – original draft:** Emma S. Cowley, Jekaterina Schneider.

**Writing – review & editing:** Emma S. Cowley, Jekaterina Schneider.

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
