## [Decision Letter · Decision Letter 0]

4 Dec 2024

PONE-D-24-25040“I Sometimes Feel Like I Can’t Win!”: An Exploratory Mixed-Methods Study of Women’s Body Image and Experiences of Exercising in Gym SettingsPLOS ONE

Dear Dr. Cowley,

Thank you for submitting your manuscript to PLOS ONE. After careful consideration, we feel that it has merit but does not fully meet PLOS ONE’s publication criteria as it currently stands. Therefore, we invite you to submit a revised version of the manuscript that addresses the points raised during the review process.

We look forward to receiving your revised manuscript.

Kind regards,

Filip Haegdorens, Ph.D

Academic Editor

PLOS ONE

Additional Editor Comments:

Dear authors,

Thank you for your patience. It was very difficult to secure reviewers for this interesting paper.

Please provide an answer to the remarks of the two reviewers. I believe that if all remarks can be addressed, this manuscript could be accepted for publication.

Reviewers' comments:

Reviewer's Responses to Questions

**Comments to the Author**

1. Is the manuscript technically sound, and do the data support the conclusions?

Reviewer #1: Partly

Reviewer #2: Yes

2. Has the statistical analysis been performed appropriately and rigorously? 

Reviewer #1: Yes

Reviewer #2: I Don't Know

3. Have the authors made all data underlying the findings in their manuscript fully available?

Reviewer #1: Yes

Reviewer #2: Yes

4. Is the manuscript presented in an intelligible fashion and written in standard English?

Reviewer #1: Yes

Reviewer #2: Yes

5. Review Comments to the Author

Reviewer #1: Thank you for your research and article. I very much enjoyed reviewing this paper.

Some notes I have before publishing:

Change: 'survey was piloted and before data collection.'- delete 'and'

Please define the 'reCAPTCHA test' somewhere in your text. The subheading 'procedures' seems suited. Is there any evidence that this would exclude certain potential responders? (i.e. dyslexia, ...) making it less inclusive? ( for discussion)

It seems strange to me that for the demographics you've presented almost all have 24% missing? was there an issue with these survey questions?

I'm struggling to understand what table 3 adds to your results, discussion and conclusion. I would elaborate on the found results here and your qualitative results. Also I would use the actual name of the variables in the first row instead of 1,2,3,4,5 - it is quite confusing and difficult to read this way.

for subtitle 'Judgement about appearance' - the participants quote addresses something other than 'body shape' while most of the subtheme does refer to the shape of a woman (manly, muscular, thin, small, ...). It seems that the quote you used revealed something else i.e a woman not worrying about her shape, make up or hair, BUT her skin. The first paragraph should elaborate more on what issues women often refer to as 'appearance' (skin, hair, ...).

The subtheme 'Lack of Clothes that Fit' does not fully cover the meaning of the paragraphs below. The participants quotes also include other issues with gym clothing (e.g. sweatmarks, seeing underwear through leggings, ...) that doesn't really refer to the size or fit, but rather the materials gym clothes are made of.

I think it's strange that in the introduction and your questioning you kind of hypothesize that mirrors are a problem, while in the discussion you do not address that participants in this study (who predominantly like strength based exercise -> often with weights, and regularly attend the gym) do not really care or even like the mirrors. It could be important to address this and it reveals a contradicting result with other literature.

In the recommendations I'm missing the positive and empowering messages, i.e. there are environments that empower women, what do they look like and what can we learn from them?  maybe also recommend research on the specific factors that make a gym more inclusive to women and 'external factors'- i.e. not gym or attire related (in discussion) ?

Your study is very focused on the negative, while most of these participants attend the gym a lot (only 3 gym goers less than 1h a week) - what about those women not attending the gym? or attending the gym less than they would like to? --- please further explain why you chose to include both past gym go-ers, relatively low frequency 'gym-members' and frequent gym go-ers with regard to your research questions.

Reviewer #2: I will preface this review by saying I am not an expert in quant analysis, so will be assuming the authors have relayed these portions accurately and followed journal policy where relevant. I instead will focus my review on the qualitative aspects and how it fits with the existing literature. Hopefully the editors can check to make sure they are happy with the quant portions.

I absolutely think this work should be published. Whilst some of the themes have already been established in the existing literature, to my knowledge this is the largest sample that has been surveyed regarding this topic, and the findings offer clear contribution beyond the themes that have already been discussed elsewhere. The authors do a fantastic job of situating their findings within this existing literature, and noting where their findings contribute to or build upon this literature, as well as potential contradictions with prior research. This is a clear contribution to knowledge.

Overall I thought the piece was really well written and the authors did a fantastic job of highlighting key themes with relevant quotes and examples. I had a couple of queries, but I do not think the authors necessarily need to address these in order for the article to be published, they were just general curiosities.

The authors say the survey was open worldwide but note that the sample often had a similar background upon analysis. I’m interested in whether British women’s experiences significantly varied from the international portion of the sample, and whether there was any data on these cultural differences? With that said, potentially this is planned for a future paper, so I don’t think it needs to be addressed unless the authors want to add something on it.

With the finding that most women were not concerned about becoming “too” muscular seemingly contrasting with prior literature, do the authors have a theory for why their findings may differ? Is it the rise of social media fitspiration culture normalising women with more muscular body types, or do you not feel the data offers enough info on this to say? Just something I would be interested to hear their thoughts on, but by all means publish the article without this.

As a final note, I found it interesting the authors received 'bot' responses and had to filter for these. I would be curious knowing a little more about this filtering process, but again do not think this is necessarily needed in the current paper (since it is more of a side issue on methods).

6. PLOS authors have the option to publish the peer review history of their article (what does this mean?). If published, this will include your full peer review and any attached files.

Reviewer #1: No

Reviewer #2: No

---

## [Author Response · Author response to Decision Letter 0]

6 Dec 2024

REVIEWER #1: 

Thank you for your research and article. I very much enjoyed reviewing this paper. Some notes I have before publishing:

RESPONSE: We want to thank the Reviewer for their time and their constructive and insightful feedback on our manuscript. We have addressed each comment point by point below and made the relevant changes to the manuscript using yellow-highlighted font. We believe the manuscript has substantially improved as a result of these revisions and look forward to the next steps.

Change: 'survey was piloted and before data collection.'- delete 'and'

RESPONSE Thank you for spotting this issue, we have now deleted “and” from this sentence (please see page 6). We have also conducted a thorough grammar and spell check of the entire manuscript. 

Please define the ‘reCAPTCHA test’ somewhere in your text.

RESPONSE We have provided a definition of “reCAPTCHA” on page 10 of the revised manuscript.

Additionally, we have now specified that our procedure for screening bots and fraudulent responses was based on Qualtrics data quality screening procedures (https://www.qualtrics.com/support/survey-platform/survey-module/survey-checker/fraud-detection/) and recently published guidelines for detecting fraudulent responses in web-based research, to provide additional context for the reader (please see page 10).

Schneider, J., Ahuja, L., Dietch, J. R., Folan, A. M., Coleman, J., & Bogart, K. (2024). Addressing fraudulent responses in quantitative and qualitative internet research: Case studies from body image and appearance research. Ethics & Behavior, 1–13. https://doi.org/10.1080/10508422.2024.2411400

The subheading 'procedures' seems suited. Is there any evidence that this would exclude certain potential responders? (i.e. dyslexia, ...) making it less inclusive? (→ for discussion)

RESPONSE Thank you for raising this important point.

The survey was designed with inclusivity and accessibility in mind. Questions were written in lay terms to ensure clarity for the general population of women aged 18+ without subject matter expertise. 

To further ensure accessibility, the survey was assessed using the Qualtrics accessibility tool, which confirmed that the questions were WCAG 2.0 AA compliant. While the tool flagged ‘minor’ concerns for participants with cognitive or learning disabilities, the only questions that received a ‘fair’ accessibility rating were Likert Scale items. These items were drawn from validated psychosocial questionnaires and could not be modified without compromising their validity.

It seems strange to me that for the demographics you’ve presented almost all have 24% missing? Was there an issue with these survey questions?

RESPONSE Thank you for raising this issue. There are two plausible explanations for the missing data on the demographic questions. First, demographic questions were presented at the end of the survey, to reduce participants’ cognitive load and prioritise other survey sections. 

Second, in our survey, we made it clear to participants that the demographic questions were optional. This was to ensure participants felt encouraged to remain anonymous if they chose to do so, given the sensitive nature of some of the questions in the survey (e.g., questions related to sexual harassment in the gym). 

It is likely that individuals who chose not to complete the demographic section skipped all the questions, rather than selecting which ones to answer and which ones to not respond to. 

I’m struggling to understand what Table 3 adds to your results, discussion and conclusion. I would elaborate on the found results here and your qualitative results. Also I would use the actual name of the variables in the first row instead of 1,2,3,4,5 – it is quite confusing and difficult to read this way.

RESPONSE To streamline the manuscript and improve readability, we chose to summarise the results of the correlation analyses in the text, rather than providing individual statistical values for all correlations. Therefore, Table 3 provides the statistical values of all the correlations between the psychosocial variables included in the manuscript. 

Including correlational statistics provides transparency and allows readers to objectively evaluate the strength and direction of relationships, supporting reproducibility and alignment with scientific standards. It also enables comparison with existing literature and informs future research.

However, we are happy to defer to the Editor regarding whether or not to retain Table 3. 

Finally, in line with the Reviewer’s comment, we have changed the first row of Table 3 to the variable names, rather than numbers. 

For subtitle ‘Judgement about appearance’ – the participants quote addresses something other than 'body shape' while most of the subtheme does refer to the shape of a woman (manly, muscular, thin, small, ...). It seems that the quote you used revealed something else i.e a woman not worrying about her shape, make up or hair, BUT her skin. The first paragraph should elaborate more on what issues women often refer to as 'appearance' (skin, hair, ...). 

RESPONSE Thank you for your helpful comment. We agree that the sub-theme introduction sentence needed refinement to better align with the corresponding participant quote and fully capture the diversity of concerns expressed. We have now amended the text to read:

“Appearance-related concerns were varied and complex, encompassing not only feelings of self-consciousness regarding physical body shape and size, but also discomfort related to hair, skin issues, and other concerns.” (please see page 11).

The subtheme ‘Lack of Clothes that Fit’ does not fully cover the meaning of the paragraphs below. The participants’ quotes also include other issues with gym clothing (e.g. sweatmarks, seeing underwear through leggings, ...) that doesn't really refer to the size or fit, but rather the materials gym clothes are made of.

RESPONSE We agree with the Reviewer and have changed the subtheme name to “Clothing Challenges” to better represent women’s concerns with gym clothing (please see page 14). 

I think it's strange that in the introduction and your questioning you kind of hypothesize that mirrors are a problem, while in the discussion you do not address that participants in this study (who predominantly like strength based exercise → often with weights, and regularly attend the gym) do not really care or even like the mirrors. It could be important to address this and it reveals a contradicting result with other literature. 

RESPONSE Thank you for this feedback and we agree that our hypothesis versus participant feedback was opposing in regards to mirrors. Following from this study, we are currently conducting a follow-up interview study with participants who took this study to better understand this contradiction and take a deeper dive into some of the other dichotomies within the results (e.g., muscularity ideals).

Further, we have added section ‘4.4 Future Directions’ in the revised manuscript, where we provide a concise overview of where we recommend future research should focus:

“Future research should employ more targeted recruitment strategies to ensure the inclusion of underrepresented groups, such as women from non-WEIRD countries, those from the LGBTQIA+ community, and women with disabilities. Research should also explore the experiences of women who have discontinued gym-based exercise to gain a better understanding into the factors contributing to their drop-off. Building on the findings of this study, follow-up research is also required to understand the contradictory results highlighted in the data. For example, the majority of participants reported feeling neutral or positive about exercising in front of mirrors, which runs contrary to much of the existing literature. Further, investigation is also warranted into the evolving muscularity ideals and the shifting perceptions of the ‘ideal’ female body, as these cultural factors likely influence women’s experiences and engagement in gym settings. To this end, we are currently conducting a follow-up qualitative study to gain greater insights into these findings.” (please see page 23).

In the recommendations I’m missing the positive and empowering messages, i.e. there are environments that empower women, what do they look like and what can we learn from them? → maybe also recommend research on the specific factors that make a gym more inclusive to women and ‘external factors’ – i.e. not gym or attire related (in discussion)?

RESPONSE Thank you for this feedback. In response, we have added section ‘4.3 Implications’ in the revised manuscript, where we provide a short summary of our multi-level recommendations based on Figure 2. This section focuses on actionable facilitators highlighted by participants in the survey, as well as prior research, detailing ways in which women feel empowered to exercise in gym spaces: 

“Building on our findings and prior research, we outline important implications for creating safe, inclusive, and empowering gym spaces for women at the individual, interpersonal, organisational, and societal levels (see Figure 2). For example, individuals should consider focusing on non-appearance related goals, emphasising the physical, mental, and emotional health benefits of exercise. To foster safer interpersonal relations, gyms should enforce comprehensive anti-harassment policies and provide staff training to cultivate supportive and inclusive environments. Gym layouts should be designed to prioritise accessibility, ensuring that spaces and equipment accommodate diverse needs and enhance users’ comfort and confidence. Finally, public health campaigns should promote a cultural shift by celebrating exercise as a means to improve health and well-being, moving away from a sole focus on aesthetics.” (please see page 23)

Your study is very focused on the negative, while most of these participants attend the gym a lot (only 3 gym goers less than 1h a week) – what about those women not attending the gym? or attending the gym less than they would like to? – please further explain why you chose to include both past gym go-ers, relatively low frequency 'gym-members' and frequent gym go-ers with regard to your research questions.

RESPONSE The aim of our study was to gain experience from current and past gym-goers, to identify distinct barriers and motivators to why some women continue going to the gym regularly, while others drop out. We have now added the following information to our Methods section to better justify this decision: 

“The study aimed to explore the experiences of both active and inactive women to identify the unique barriers and motivators influencing gym attendance and behaviours. Including both groups allowed for a comprehensive understanding of the factors that sustain regular gym participation and those that contribute to dropout.” (please see page 5)

However, given the topic of the study, we naturally got mostly active women in our sample. We have acknowledged this in our Limitations section as follows: 

“Despite the diverse geographical representation, most participants were White, heterosexual, identified as “normal weight” and able-bodied, were current gym-goers, and lived in western, educated, industrialised, rich, and democratic (WEIRD) countries. This limits the applicability of our findings to women who no longer engage in exercise (particularly in a gym setting) or those of diverse ethnicities, body sizes, sexualities, and abilities.” (please see page 21)

We have also included this as a recommendation for future research in our new Future Directions section: 

“Research should also explore the experiences of women who have discontinued gym-based exercise to gain a better understanding into the factors contributing to their drop-off.” (please see page 23)

REVIEWER #2: 

I will preface this review by saying I am not an expert in quant analysis, so will be assuming the authors have relayed these portions accurately and followed journal policy where relevant. I instead will focus my review on the qualitative aspects and how it fits with the existing literature. Hopefully the editors can check to make sure they are happy with the quant portions.

I absolutely think this work should be published. Whilst some of the themes have already been established in the existing literature, to my knowledge this is the largest sample that has been surveyed regarding this topic, and the findings offer clear contribution beyond the themes that have already been discussed elsewhere. The authors do a fantastic job of situating their findings within this existing literature, and noting where their findings contribute to or build upon this literature, as well as potential contradictions with prior research. This is a clear contribution to knowledge.

Overall I thought the piece was really well written and the authors did a fantastic job of highlighting key themes with relevant quotes and examples. I had a couple of queries, but I do not think the authors necessarily need to address these in order for the article to be published, they were just general curiosities.

RESPONSE: We want to thank the Reviewer for their time and their constructive and insightful feedback on our manuscript. We have addressed each comment point by point below and made the relevant changes to the manuscript using yellow-highlighted font. We believe the manuscript has substantially improved as a result of these revisions and look forward to the next steps.

The authors say the survey was open worldwide but note that the sample often had a similar background upon analysis. I’m interested in whether British women’s experiences significantly varied from the international portion of the sample, and whether there was any data on these cultural differences? With that said, potentially this is planned for a future paper, so I don’t think it needs to be addressed unless the authors want to add something on it.

RESPONSE Thank you for this thoughtful comment. We agree that, although the survey was open internationally, the majority of our participants were British and all respondents came from WEIRD countries. Therefore, we did not conduct analyses based on cultural differences. However, we acknowledge the importance of examining cultural variation in gym experiences and agree that this represents an important avenue for future research. 

To address this, we have highlighted this as a limitation of our study, acknowledging the homogeneity of our sample (please see page 22).

We have also added a sentence in section ‘4.4 Future Directions’, emphasising the need for research that includes seldom-heard groups, particularly women from non-WEIRD countries (please see page 23).

With the finding that most women were not concerned about becoming “too” muscular seemingly contrasting with prior literature, do the authors have a theory for why their findings may differ? Is it the rise of social media fitspiration culture normalising women with more muscular body types, or do you not feel the data offers enough info on this to say? Just something I would be interested to hear their thoughts on, but by all means publish the article without this.

RESPONSE Thank you for your thoughtful comment regarding the finding that most women in our study were not concerned about becoming “too” muscular and how this contrasts with prior literature. We agree that this may reflect the growing influence of social media and the normalisation of more muscular body types through #Fitspiration and similar content, which promotes the “fit” ideal as aspirational. While our data does not provide direct evidence for this cultural shift, it aligns with existing literature suggesting that appearance ideals are evolving from thinness to fitness. We have now added further elaboration of this in our Discussion section: 

“The rise of the muscularity ideal among women appears to reflect a broader shift in appearance ideals from thinness to fitness, often characterised by lean, toned, and muscular physiques. This trend is influenced by the proliferation of social media content, particularly #Fitspiration, which promotes “strong” and “fit” as desirable traits. This cultural shift could explain the higher internal

---

## [Editor Report · Decision Letter 1]

16 Dec 2024

“I Sometimes Feel Like I Can’t Win!”: An Exploratory Mixed-Methods Study of Women’s Body Image and Experiences of Exercising in Gym Settings

PONE-D-24-25040R1

Dear Dr. Cowley,

We’re pleased to inform you that your manuscript has been judged scientifically suitable for publication and will be formally accepted for publication once it meets all outstanding technical requirements.

Kind regards,

Filip Haegdorens, Ph.D

Academic Editor

PLOS ONE

---

## [Editor Report · Acceptance letter]

22 Dec 2024

PONE-D-24-25040R1 

PLOS ONE

Dear Dr. Cowley, 

I'm pleased to inform you that your manuscript has been deemed suitable for publication in PLOS ONE. Congratulations! Your manuscript is now being handed over to our production team.

Kind regards, 

on behalf of

prof. dr. Filip Haegdorens 

Academic Editor

PLOS ONE